# Neoadjuvant camrelizumab plus nab-paclitaxel and epirubicin in early triple-negative breast cancer: a single-arm phase II trial

Chengzheng Wang[1], Zhenzhen Liu [1]✉, Xiuchun Chen[1], Jianghua Qiao[1], Zhenduo Lu[1], Lianfang Li[1], Xianfu Sun[1], Chongjian Zhang[1], Xiayu Yue[1], Qingxin Xia[2], He Zhang[2] & Min Yan [1]

Immunotherapy combined with chemotherapy has been demonstrated to be effective in early triple-negative breast cancer (TNBC). In this single-arm, phase II study with Simon's two-stage design, we investigated the efficacy and safety of neoadjuvant camrelizumab plus chemotherapy in patients with early TNBC (NCT04213898). Eligible female patients aged 18 years or older with histologically confirmed treatment-naïve early TNBC were treated with camrelizumab (200 mg, on day 1), nab-paclitaxel (125 mg/m², on days 1, 8, and 15), and epirubicin (75 mg/m², on day 1) every three weeks for six cycles. The primary end point was the pathological complete response; secondary endpoints included safety, objective response rate, and long-term survival outcomes of event-free survival, disease-free survival, and distant disease-free survival. A total of 39 patients were enrolled between January 2020 and October 2021. Twenty-five patients achieved a pathological complete response (64.1%, 95%CI: 47.2, 78.8). The objective response rate was 89.7% (95%CI: 74.8, 96.7), including 35 patients with partial responses. Treatment-related adverse events of grade 3 or 4 occurred in 30 (76.9%) patients. In conclusion, the trial meets the pre-specified endpoints showing promising efficacy and manageable safety of neoadjuvant camrelizumab plus nab-paclitaxel and epirubicin chemotherapy in female patients with early TNBC. Long-term survival outcomes are still pending.

Triple-negative breast cancer (TNBC) is a heterogeneous disease characterized by lack of or minimal expression of estrogen receptor (ER) and progesterone receptor (PR) and lack of overexpression and/ or amplification of human epidermal growth factor receptor 2 (HER2). It accounts for almost 15%–20% of all breast cancer subtypes and is more aggressive than other subtypes, with a higher risk of recurrence, early metastases, and developing drug resistance[1,2]. Cytotoxic chemotherapy remains the backbone of the standard systemic treatment for TNBC due to the lack of specific targets. For patients with high-risk early TNBC, neoadjuvant chemotherapy remains a major choice[3,4]. Neoadjuvant therapy could not only render inoperable tumors operable but also facilitate breast-conservation. Moreover, it provides an

[1]Department of Breast Disease, Henan Breast Cancer Center, The Affiliated Cancer Hospital of Zhengzhou University & Henan Cancer Hospital, Zhengzhou 450008, China. [2]Department of Pathology, The Affiliated Cancer Hospital of Zhengzhou University & Henan Cancer Hospital, Zhengzhou 450008, China. ✉e-mail: zlyyliuzhenzhen0800@zzu.edu.cn

excellent research platform to rapidly test drug efficacy and explore potential biomarkers, thus guiding the subsequent treatment decision-making. Following neoadjuvant therapy, patients who achieve a pathological complete response (pCR, ypT0ypN0 or ypT0/isypN0) have improved survival, particularly those with more aggressive breast cancer subtypes like TNBC[5]. Those with a residual disease could be further treated with adjuvant therapy, yet approximately 20−30% of the patients developed recurrent diseases and died[6,7].

Immune checkpoint inhibitors (ICIs) have been established as a part of the standard of care for treating breast cancer. In patients with early TNBC, the KEYNOTE-522 and IMpassion031 trials demonstrated a higher pCR with neoadjuvant pembrolizumab (an anti-progression cell death 1 [PD-1] inhibitor) or atezolizumab (an anti-programmed death ligand-1 [PD-L1] inhibitor) when combined with chemotherapy[8,9]. Based on data from the KEYNOTE-522 trial, pembrolizumab combined with neoadjuvant chemotherapy has been recommended for high-risk early TNBC. Neoadjuvant ICIs plus chemotherapy have been reshaping the treatment landscape of early TNBC. Camrelizumab is a humanized monoclonal antibody that specifically binds to PD-1 and blocks its interaction with PD-L1, thus restoring T-cell activation and reversing tumor immune evasion. Camrelizumab plus anti-angiogenic agents (apatinib or famitinib) with or without single-agent chemotherapy (nab-paclitaxel or eribulin) have been reported to be effective and safe in the treatment of advanced TNBC[10−12].

Anthracycline combined with taxanes is a common chemotherapy regimen for the treatment of TNBC. However, there is a lack of research on combining ICIs with anthracycline and taxanes in the neoadjuvant setting. Herein, we conducted this single-arm phase II study to evaluate the efficacy and safety of neoadjuvant camrelizumab plus nab-paclitaxel and epirubicin chemotherapy in patients with early TNBC. We reported here the perioperative outcomes of neoadjuvant camrelizumab plus chemotherapy. The trial is still ongoing and being followed up.

In this study, neoadjuvant therapy with camrelizumab plus nab-paclitaxel and epirubicin results in a pCR rate of 64.1%, with a manageable safety profile. This study supports the feasibility of study treatment in early TNBC.

## Results

### Patient characteristics

Between January 3, 2020 and October 25, 2021, 97 patients were screened for eligibility, and 39 were enrolled (Fig. 1). All patients received neoadjuvant camrelizumab plus chemotherapy, and eight patients prematurely discontinued the treatment due to adverse events (n = 7) or disease progression (n = 1). All patients underwent surgery and were evaluable for pathological tumor response.

The median age of patients was 46 (range: 31−59) years. Most patients were premenopausal (84.6%) and had T2 (92.3%) and N0-1 (84.6%) diseases. Tumor PD-L1 expression was assessed in 34 (87.2%) patients, and 12 (30.8%) patients were positively stained. The baseline patient characteristics are summarized in Table 1.

### Efficacy

In the first stage, eight (88.9%) of the nine patients initially enrolled achieved a pCR, exceeding the threshold required for proceeding to the second stage of enrollment. Among 39 patients finally enrolled, 25

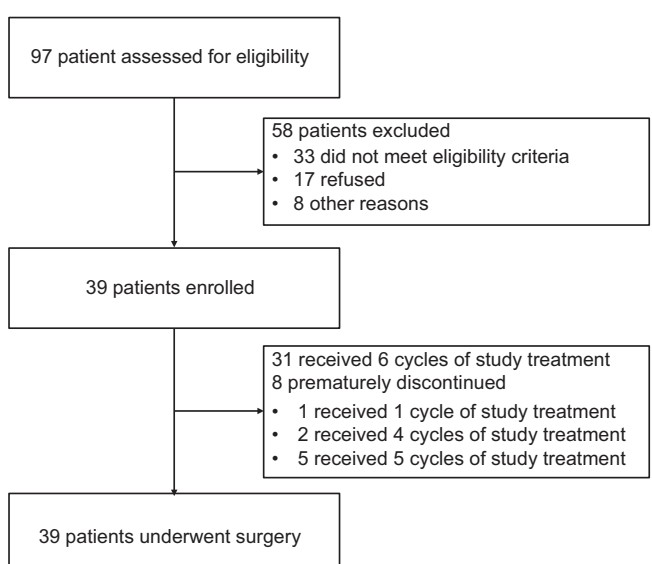

**Fig. 1 | Trial profile.** A total of 97 patients were screened for eligibility, 58 were excluded, and finally 39 were enrolled. All 39 patients received at least one dose of the study treatment and proceeded to surgical resection. All enrolled patients were accessible for efficacy and safety analyses.

**Table 1 | Baseline patient characteristics**

|  | Total (n = 39) |
|---|---|
| Age, years | |
| Median (IQR) | 46 (31, 59) |
| ≤40 | 13 (33.3) |
| >40 | 26 (66.7) |
| Menopausal status | |
| Premenopausal | 33 (84.6) |
| Menopause | 6 (15.4) |
| ECOG performance status, n (%) | |
| 0 | 39 (100.0) |
| 1 | 0 (0.0) |
| ER status, n (%) | |
| <1 | 29 (65.5) |
| 1–10 | 10 (34.5) |
| HER2 status score, n (%) | |
| IHC 0 | 18 (46.2) |
| IHC ≥ 1+ | 21 (53.8) |
| Ki-67 level, n (%) | |
| ≤30% | 2 (5.1) |
| >30% | 37 (94.9) |
| Clinical T stage, n (%) | |
| T1 | 2 (5.1) |
| T2 | 36 (92.3) |
| T3 | 1 (2.6) |
| Clinical N stage, n (%) | |
| N0 | 17 (43.6) |
| N1 | 16 (41.0) |
| N2 | 3 (7.7) |
| N3 | 3 (7.7) |
| Overall clinical stage, n (%) | |
| Stage II | 32 (82.1) |
| Stage III | 7 (17.9) |
| PD-L1 status, n (%)[a] | |
| Positive | 12 (30.8) |
| Negative | 23 (59.0) |
| Unknown | 4 (10.2) |

IQR interquartile range, ECOG eastern cooperative oncology group, HER2 human epidermal growth factor receptor 2.
[a]PD-L1 positive is defined as a combined positive score of ≥1.

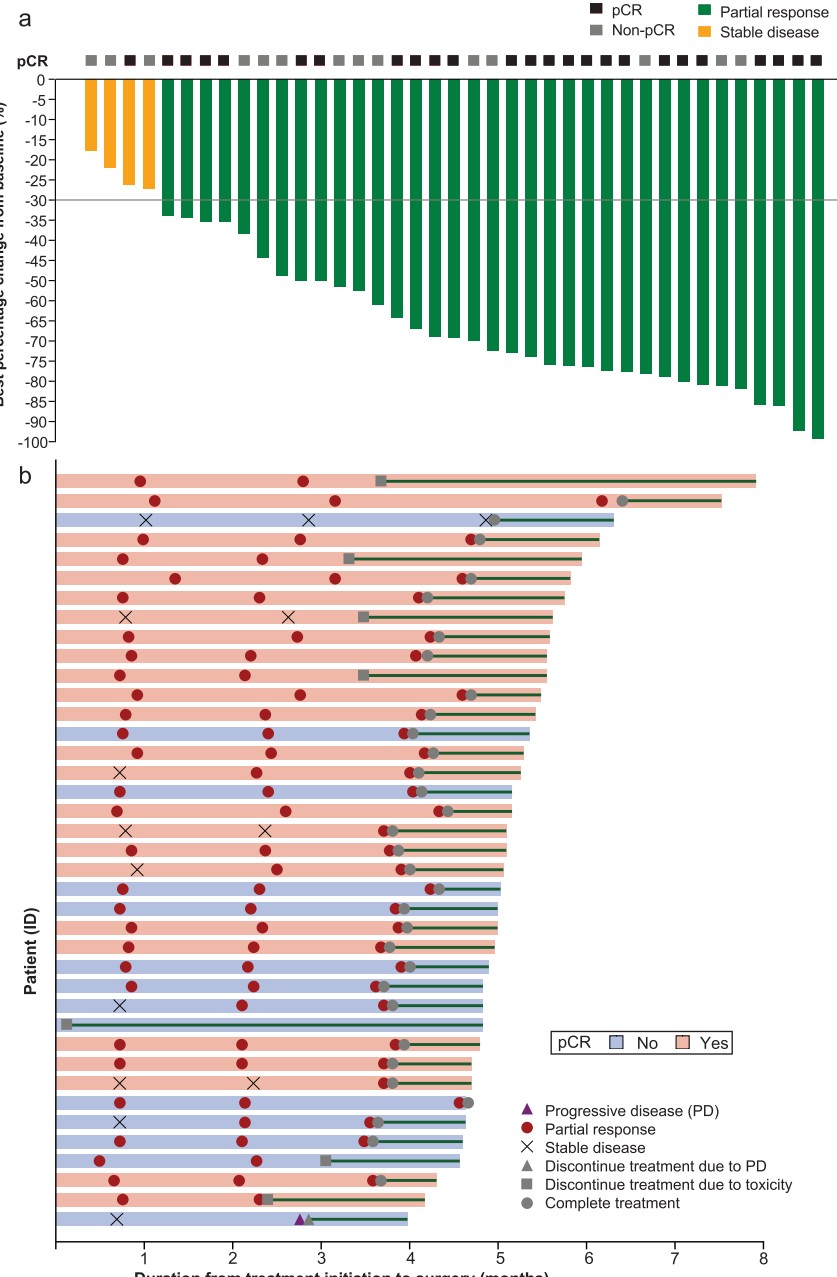

**Fig. 2 | Clinical and pathological response to neoadjuvant camrelizumab plus chemotherapy in 39 patients with early triple-negative breast cancer. a** The waterfall plot showing the best percentage changes in tumor size of target lesions from baseline. Green bars indicate patients having the best overall response of partial response, and yellow bars indicate those having stable disease. The black boxes above indicate patients achieving a pathological complete response (pCR), while gray boxes indicate those without. The gray horizontal line indicates a 30% decrease from the baseline. **b** The swimmer plot showing the individual patient's journey from treatment initiation to surgery. The length of the bar indicates the duration from treatment initiation to surgery, and the colors indicate the pathological tumor response (light pink: pCR, baby blue: non-pCR). Tumor response at each assessment is shown as a red circle (partial response), gray cross (stable disease), or purple triangle (progressive disease). The reason for the end of the treatment is shown as a gray triangle (discontinuation due to disease progression), square (discontinuation due to toxicity), or circle (completion). Green lines indicate the duration from the end of the treatment to surgery. Source data are provided as a Source Data file.

(64.1%, 95%CI: 47.2, 78.8) achieved a pCR (ypT0/Tis ypN0). A total of 35 patients achieved a partial response, and four had stable disease, resulting in an objective response rate of 89.7% (95%CI: 74.8, 96.7) (Fig. 2a). Interestingly, one patient had a radiological stable disease while achieving a pCR following neoadjuvant therapy with camrelizumab plus chemotherapy.

Among the 31 patients who completed six cycles of neoadjuvant therapy, 20 (64.5%) had a pCR. Seven patients prematurely discontinued study treatment due to adverse events, including two discontinued camrelizumab only and five discontinued both camrelizumab and chemotherapy. Five (71.4%) of the seven patients who discontinued treatment achieved a pCR when proceeding to surgery (Fig. 2b). Among these five patients, four received four cycles, and one received five cycles of neoadjuvant therapy (Supplementary Table 1). The other two patients who did not achieve a pCR received one and five cycles of neoadjuvant therapy, respectively.

The percentages of patients with a pCR were 91.7% (11/12) in the PD-L1-positive patients and 40.9% (9/22) in the PD-L1-negative patients

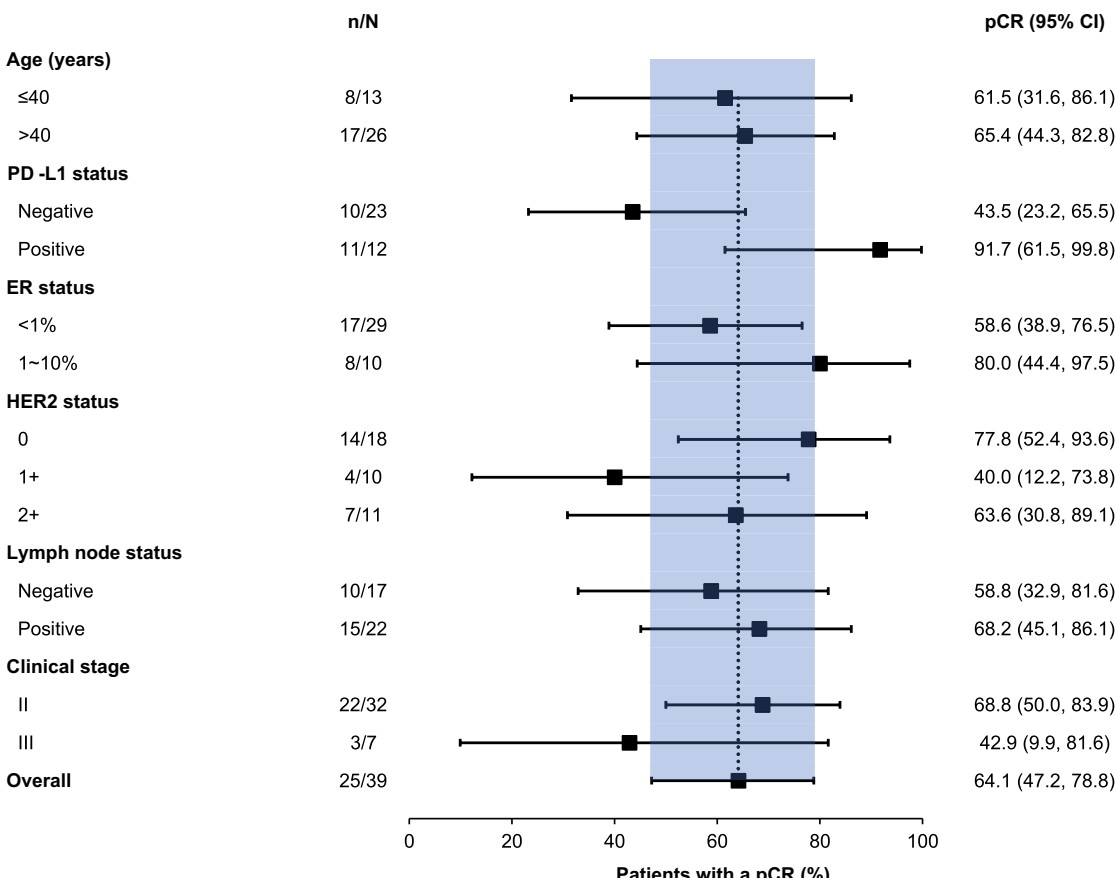

**Fig. 3 | Subgroup analysis of pCR.** Data are presented as percentage and 95% confidential interval (CI). The *n* indicates the number of patients achieving a pCR; *N* indicates the total number of patients available for pathological assessment. The black vertical dotted line indicates the pathological complete response rate in the overall population, and the baby blue area shows the corresponding 95% CI. The black squares indicate the pCR for each subgroup of patients, while the length of black horizontal lines indicates the corresponding 95% CI. Source data are provided as a Source Data file.

(difference: 48.2%, 95% CI: 22.6%, 73.8%). The post-hoc subgroup analysis showed that pCR rates were generally consistent across other subgroups (Fig. 3).

**Safety**

All patients experienced at least one treatment-related adverse event. The Grade 3 or 4 treatment-related adverse events occurred in 76.9% (30/39) of patients, with white blood cell decreased (56.4%), neutropenia (41.0%), and anemia (20.5%) being the most common. Serious treatment-related adverse events occurred in five (12.8%) patients. No treatment-related death occurred. The common treatment-related adverse events occurring in ≥ 10% of patients are shown in Table 2.

Immune-related adverse events of any grade occurred in 24 (61.5%) patients, while those of grade 3 or 4 occurred in seven (17.9%) patients. The most common immune-related adverse event was reactive cutaneous capillary endothelial proliferation (17/39, 43.6%); all were grade 1 or 2 (Table 3). All seven patients who discontinued study treatment were due to immune-related adverse events.

**Discussion**

In this single-arm phase II trial, neoadjuvant camrelizumab plus nab-paclitaxel and epirubicin achieved a pCR rate of 64.1% in patients with early TNBC. Treatment-related adverse events of grade 3 or 4 occurred in 76.9% of patients. No grade 5 treatment-related adverse events occurred. Our study meets the primary endpoint showing promising anti-tumor activity with a manageable safety profile of neoadjuvant camrelizumab plus chemotherapy in patients with early TNBC.

ICIs combination strategies, including chemoimmunotherapy, have been increasingly investigated in breast cancer, given the promising anti-tumor activity of ICIs monotherapy in a limited subset of patients. Chemotherapy may induce immunogenic cell death, promote antigen release and presentation, and selectively inhibit or deplete immunosuppressive cells to stimulate anticancer immunity[13]. Besides, chemotherapy increased PD-L1 expression in tumor-infiltrating immune cells, which was explored as a potential biomarker for patient selection and prognostic prediction[14,15]. Anthracyclines have long been established as a cornerstone for the treatment of breast cancer that can stimulate immunogenicity and induce tumor cell death alongside the direct cytotoxic effects[16]. Short-term induction with doxorubicin and cisplatin created a favorable tumor microenvironment and enhanced clinical response to PD-1 inhibitors in TNBC[16]. However, steroid premedication for cytotoxic chemotherapy may impair the ICIs-induced antitumor immune response[17]. Accordingly, nab-paclitaxel that requires no steroid premedication may be an ideal candidate chemotherapeutic agent in ICIs combination strategies[18,19]. In the present study, nab-paclitaxel and epirubicin were chosen to enhance the anti-tumor activity of camrelizumab.

The pCR benefit observed in this study was generally consistent with that of neoadjuvant pembrolizumab plus platinum-containing chemotherapy reported in the KEYNOTE-522 (64.8%) trial[8] and that of neoadjuvant atezolizumab plus sequential nab-paclitaxel and anthracycline-based chemotherapy in the IMpassion031 (58%) trial[9]. However, the data seem a little bit higher than that of neoadjuvant durvalumab plus sequential nab-paclitaxel and anthracycline-based

**Table 2 | Treatment-related adverse events occurring in ≥10% of patients**

| | Total (*n* = 39) | |
|---|---|---|
| | Any grade | Grade ≥ 3 |
| Treatment-related adverse events, *n* (%) | | |
| Anemia | 35 (89.7) | 8 (20.5) |
| Leukopenia | 31 (79.5) | 22 (56.4) |
| Neutropenia | 24 (61.5) | 16 (41.0) |
| Peripheral neuropathy | 21 (53.8) | 0 (0) |
| Vomiting | 20 (51.3) | 2 (5.1) |
| Nausea | 20 (51.3) | 0 (0) |
| Hypoalbuminemia | 16 (41.0) | 0 (0) |
| ALT increased | 16 (41.0) | 1 (2.6) |
| Skin hyperpigmentation | 15 (38.5) | 0 (0) |
| Hypertriglyceridemia | 15 (38.5) | 0 (0) |
| Fatigue | 14 (35.9) | 1 (2.6) |
| Anorexia | 13 (33.3) | 0 (0) |
| Hyponatremia | 12 (30.8) | 2 (5.1) |
| Diarrhea | 12 (30.8) | 1 (2.6) |
| Rash | 11 (28.2) | 0 (0) |
| AST increased | 11 (28.2) | 2 (5.1) |
| LDH increased | 10 (25.6) | 0 (0) |
| Hypermagnesemia | 10 (25.6) | 0 (0) |
| Oral mucositis | 10 (25.6) | 0 (0) |
| Pruritus | 8 (20.5) | 0 (0) |
| Lymphocytopenia | 7 (17.9) | 1 (2.6) |
| Dry mouth | 7 (17.9) | 0 (0) |
| Cough | 7 (17.9) | 0 (0) |
| Cholesterol high | 7 (17.9) | 0 (0) |
| Hypercalcemia | 7 (17.9) | 0 (0) |
| ALP increased | 6 (15.4) | 0 (0) |
| Pyrexia | 6 (15.4) | 0 (0) |
| Platelet count decreased | 5 (12.8) | 1 (2.6) |
| Hyperglycemia | 5 (12.8) | 0 (0) |
| Hypocalcemia | 5 (12.8) | 0 (0) |
| Hyperuricemia | 5 (12.8) | 0 (0) |
| Somnolence | 5 (12.8) | 0 (0) |
| Hypokalemia | 4 (10.3) | 1 (2.6) |
| Hyperchloremia | 4 (10.3) | 0 (0) |
| Constipation | 4 (10.3) | 0 (0) |
| Headache | 4 (10.3) | 0 (0) |
| Abdominal pain | 4 (10.3) | 0 (0) |

*AST* aspartate transaminase, *LDH* lactate dehydrogenase, *ALT* alanine transaminase.

**Table 3 | Immune-related adverse events**

| | Total (*n* = 39) | |
|---|---|---|
| | Any grade | Grade ≥ 3 |
| Immune-related adverse events, *n* (%) | | |
| RCCEP | 17 (43.6) | 0 (0) |
| Hyperthyroidism | 6 (15.4) | 1 (5.1) |
| Hypothyroidism | 3 (7.7) | 1 (5.1) |
| Adrenal insufficiency | 3 (7.7) | 0 (0) |
| Immune-related pneumonia | 2 (5.1) | 2 (5.1) |
| Pancreatitis | 1 (2.6) | 1 (2.6) |
| Immune-related hepatic insufficiency | 1 (2.6) | 1 (2.6) |
| Cystoid macular edema | 1 (2.6) | 1 (2.6) |
| Hypophysitis | 1 (2.6) | 0 (0) |

*RCCEP* reactive cutaneous capillary endothelial proliferation.

(HR)-low tumors (1% ≤ ER/PR < 10%) that do not formally meet the criteria of 'triple negative' (ER and PgR < 1%, HER2-negative) were also enrolled. These HR-low tumors are known to be biologically and clinically similar to TNBC in terms of pCR and survival outcomes and have long been proposed to be redefined as TNBC[22–25]. The inclusion of this group of patients seems to be encouraging. Our subgroup analysis showed promising antitumor activity in terms of pCR irrespective of ER expression level (<1%: 58.6%; 1–10%: 80.0%), suggesting the antitumor activity of neoadjuvant immunotherapy combined with chemotherapy in patients with HR-low tumors. The ongoing phase III trials (KEYNOTE-756 and CheckMate 7FL) will help to clarify the role of adding ICIs (pembrolizumab and nivolumab) to neoadjuvant chemotherapy for early HR-low, HER2-negative cancers[26,27].

Breast cancer has historically been considered a non-immunogenic cancer. However, increasing evidence showed PD-L1 expression, particularly in ER-negative subtypes (such as TNBC or HER2-positive ones), suggesting a potential therapeutic opportunity for anti-PD-1/PD-L1 inhibitors[28–30]. A systemic review and meta-analysis of 2546 breast cancer patients showed a PD-L1 positive rate of 21% −56%, and PD-L1 expression was suggested as a promising biomarker for selecting patients who may benefit from immunotherapy[31]. The IMpassion130 trial of atezolizumab plus nab-paclitaxel in patients with metastatic TNBC showed benefits specifically in the PD-L1 positive patients[32]. However, as for early TNBC, atezolizumab plus chemotherapy enhanced benefits in both PD-L1 positive and negative populations in the IMpassion031 trial (the pCR rate in PD-L1 positive patients: 68.8% vs. 49.3%; and in PD-L1 negative patients: 47.7% vs. 34.4%)[9]. Similarly, results from the KEYNOTE-522 trial showed a pCR benefit with neoadjuvant pembrolizumab plus chemotherapy in early TNBC regardless of PD-L1 expression (CPS < 1: 45.3% vs. 30.3%; CPS ≥ 1: 68.9% vs. 54.9%; CPS ≥ 10: 77.9% vs. 59.8%)[8]. However, the conflicting results of the significant effect of PD-L1 positivity on tumor response were observed in the GeparNuevo and NeoTRIP trials of neoadjuvant durvalumab or atezolizumab in combination with chemotherapy[20,21]. Moreover, PD-L1 expression was identified as the most significant factor for predicting pCR in the NeoTRIP trial. In this study, PD-L1-positive patients seem to benefit more from neoadjuvant camrelizumab plus chemotherapy when compared with their PD-L1-negative counterparts. The variations may be related to the differences in study drugs or inhibition pathways, disease stages, PD-L1 assays, or other factors. Nevertheless, due to the small number of patients included, future studies are still needed to explore the prognostic significance of PD-L1 expression during chemoimmunotherapy in early TNBC. Additionally, given the high costs and potential immune-related toxicities, identifying biomarkers for more precise patient selection is important.

The safety profile of neoadjuvant camrelizumab plus nab-paclitaxel and epirubicin was generally consistent with those of

chemotherapy in the GeparNuevo (53.4%) trial[20] and that of neoadjuvant atezolizumab plus concurrent carboplatin and nab-paclitaxel in the NeoTRIP (48.6%) trial[21]. Whether the variations in chemotherapy regimens may exert different antitumor effects in immunotherapy combination settings still needs further exploration. Notably, the number of patients with N3 disease enrolled varied among the studies mentioned above (KEYNOTE-522: 0/784 [0%]; IMpassion031: 10/165 [6%]; GeparNuevo: 2/88 [2%]; NeoTRIP: 19/138 [14%]). None of the three (7.7%) patients with N3 disease enrolled in this study achieved a pCR. Given the small number of patients, future studies are still needed to clarify whether this subgroup of patients can benefit from the treatment.

One thing worth noting is that a 10% cut-off was used in this study to define ER/PR negativity, that is, patients with hormone receptor

neoadjuvant nab-paclitaxel and epirubicin in early TNBC patients[33]. Adverse events observed in this study were generally consistent with the known safety profiles of platinum-free neoadjuvant chemotherapy for patients with early TNBC and with the known safety profiles of camrelizumab in early or metastatic TNBC[10–12,34]. The most common treatment-related adverse events of grade 3 or greater were leukopenia, neutropenia, and anemia, consistent with the toxic effects observed typically with chemotherapy. Platinum-free neoadjuvant chemotherapy has been associated with a lower incidence of grade 3 or 4 hematological adverse events (e.g., neutropenia, anemia, and thrombocytopenia) when compared with platinum-based one in a previous meta-analysis[35]. Our study showed a comparatively lower incidence of grade 3 or 4 neutropenia/decreased neutrophil count (41.0%) when compared with previous studies of immunotherapy in combination with platinum-based neoadjuvant chemotherapy (KEY-NOTE-522: 53.3%; NeoTRIP: 49%)[8,21]. Meanwhile, we did not find any signals of less hematological toxicity regarding anemia, leukopenia, or thrombocytopenia. However, the indirect cross-trial comparison should be interpreted with caution.

The addition of camrelizumab did not deteriorate chemotherapy-related toxicity. The immune-related adverse events occurred in almost 62% of patients; most were grade 1 or 2. Interestingly, five of the seven patients who discontinued treatment due to adverse events achieved a pCR after four or five cycles of neoadjuvant therapy. The standard neoadjuvant chemotherapy with at least four cycles was previously found to be significantly associated with favorable survival outcomes in breast cancer patients, especially in those with HER2 positive and TNBC[36]. However, the optimal treatment in the immunotherapy combination setting is largely unknown. Close monitoring of adverse events emerging during the treatment and distinguishing those immunotherapy-related are of significant importance for clinical patient management. Also, the treatment decision should be made after careful consideration of clinical and imaging findings, alongside the safety data of patients, so as to find a perfect balance between efficacy and safety. The long-term survival outcomes are still pending and warrant further investigation.

In this study, we added immunotherapy to a combination chemotherapy regimen, and the results showed promising efficacy. Another advantage of this study is that patients with HR-low tumors that are biologically and clinically similar to triple-negative tumors but do not formally meet the 'triple negative' criteria are enrolled. Some limitations must be acknowledged. This is a single-arm study with a small sample size. Nevertheless, we used the two-stage design and get statistically significant results. Besides, correlative studies of stromal tumor-infiltrating lymphocytes and tumor mutational burden were not performed in this study. Additionally, the assessment of long-term survival data and safety profiles is crucial when evaluating a potentially curative treatment. The contribution of adjuvant therapy with ICIs to treatment efficacy remains to be established, as well as the optimal treatment to be offered in cases of residual disease. Follow-up and subsequent analyses to assess overall survival and long-term safety are ongoing.

To sum up, this trial meets the prespecified endpoints showing showed promising anti-tumor activity and manageable safety of camrelizumab plus neoadjuvant nab-paclitaxel and epirubicin chemotherapy in patients with early TNBC, without compromising the surgery. The long-term survival outcomes are still pending.

## Methods
### Study design and patients
This investigator-initiated single-arm, open-label, phase II trial was conducted at the Affiliated Cancer Hospital of Zhengzhou University in China. The study was performed in compliance with the Good Clinical Practice guideline and Declaration of Helsinki (revised in 2013). The study protocol and any amendments were approved by Institutional

Review Board of the Affiliated Cancer Hospital of Zhengzhou University (Ethics number: 2019110712). All patients provided written informed consent before participation. No participation compensation was provided. The trial was registered with ClinicalTrials.gov (NCT04213898) on December 27, 2019. The first patient was enrolled on January 3, 2020, and the last was enrolled on October 25, 2021.

Eligible patients were women aged 18 years or older with an Eastern Cooperative Oncology Group (ECOG) performance status of 0–1 and histologically confirmed treatment-naïve non-metastatic (stage IIA-IIIC) TNBC, defined as ER and PR negative or low expression by immunohistochemistry (PR or ER staining in <10% of tumor cells irrespective of intensity) and HER2 negative by immunohistochemistry (1, 1+, or 2+) or fluorescence in situ hybridization (<2.0) in core biopsies. Tumor staging was performed according to the eighth edition of the American Joint Committee on Cancer TNM staging system. Other key inclusion criteria were having measurable lesions of ≥2 cm or those of 1-2 cm with lymph node involvement on ultrasound or magnetic resonance imaging (MRI), no evidence of distant metastasis (bilateral mammography, breast ultrasound, chest X-ray, or computed tomography [CT] scan; liver ultrasound, CT or MRI scan; and bone scan), and having adequate organ function.

Patients were excluded if they had inflammatory or metastatic breast cancer, other malignant diseases (except for cured cervical carcinoma in situ and basal cell carcinoma of the skin), had previously received systematic treatment (including anti-PD-1/PD-L1 inhibitors) or radiotherapy, or if they had participated in another trial, or during pregnancy or lactation. Detailed information can be found in the study protocol (see Supplementary Note in the Supplementary Information file).

### Treatments
Patients were treated with neoadjuvant camrelizumab plus chemotherapy every three-week for six cycles. Camrelizumab was administered intravenously at 200 mg on day one of every three-week cycle. Chemotherapy consisted of nab-paclitaxel at 125 mg/m$^2$ on days one, eight, and 15 and concurrent epirubicin at 75 mg/m$^2$ on day one every three-week, according to the recommendation of the NCCN Guidelines for Breast Cancer version 3. 2019 and the Guidelines of Chinese society of clinical Oncology (CSCO) on diagnosis treatment of breast cancer (2019 version). The treatment was continued until completion of the treatment, progressive disease, unacceptable toxicity, or patient withdrawal. Dose reductions in paclitaxel and epirubicin were allowed no more than two times, with a final dose of no less than 75% of the total dose. No dose adjustment of camrelizumab was planned. Prophylactic antiemetics (5-HT3 RA, NK-1 RA) were routinely administered.

Patients who either discontinued or completed the neoadjuvant therapy could undergo surgery as clinically indicated. Definitive surgery (breast-conserving surgery or mastectomy with sentinel lymph node biopsy or axillary lymph node dissection) was performed according to the guideline and institutional standard procedures. Postoperative adjuvant therapy was at the discretion of the treating physicians.

### Study endpoints and assessment
The primary endpoint was pCR, defined as the percentage of patients with no evidence of viable tumor cells in both breast and sampled lymph nodes (ypT0/Tis ypN0) (14–15). The secondary endpoints included safety, objective response rate, event-free survival, disease-free survival, and distant disease-free survival.

Tumor response was assessed according to the Response Evaluation Criteria in Solid Tumors version 1.1 (RECIST v1.1) by MRI scans at baseline and every six-week thereafter (following 2, 4, and 6 cycles of the treatment). Safety was monitored throughout the study treatment and continued for at least 30 days (90 days for serious adverse events).

Treatment-related adverse events were graded according to the Common Terminology Criteria for Adverse Events (CTCAE) version 5.0. Immune-related adverse events were defined using the Medical Dictionary for Regulatory Activities (MedDRA) preferred terms and were graded according to the National Comprehensive Cancer Network (NCCN) Guidelines for the management of toxicity related to ICIs.

Additionally, tumor samples were collected from the initial biopsy. PD-L1 expressions were determined using the PD-L1 IHC 22C3 pharmDx kit (Dako Agilent, Dako North America, Inc., Carpinteria, CA, USA). The combined positive score (CPS) was calculated as the number of PD-L1 positive tumor cells, lymphocytes, and macrophages divided by the total number of viable tumor cells multiplied by 100. A CPS score of ≥1 was considered PD-L1 positive.

### Statistical analysis

We used Simon's two-stage design. Previous studies of neoadjuvant anthracycline-taxane-based chemotherapy showed a pCR rate of 30% −40% in TNBC[18,35,37–40]. The null hypothesis of a pCR rate of 30% was adopted for neoadjuvant concurrent anthracycline and nab-paclitaxel chemotherapy in this study. Assuming a pCR rate of 55% would be achieved following neoadjuvant camrelizumab plus anthracycline and nab-paclitaxel chemotherapy, a total of 39 patients (including nine in the first stage) were required when considering a one-sided α of 5%, a power of 80%, and a drop-out rate of 10%. If at least four of nine evaluable patients achieved a pCR in the first stage, the trial would proceed to the next stage, and recruitment would be continued until a total of 39 patients were enrolled. The treatment would be considered promising if 15 or more out of 35 evaluable patients achieved a pCR.

For the endpoints of pCR and objective response rate, the point estimate and 95% confidential intervals (CIs) were summarized descriptively using the Clopper-Pearson method. Post-hoc subgroup analyses of pCR regarding patient characteristics were summarized descriptively and reported with 95% CIs (Clopper-Pearson method). Statistical analyses were performed with SAS version 9.4 software (SAS Institute Inc.) and PASS version 15.0.3 software (NSCC, LLC, Kaysville, UT, USA).

### Reporting summary

Further information on research design is available in the Nature Portfolio Reporting Summary linked to this article.

## Data availability

The study protocol is available as Supplementary Note in the Supplementary Information file. The raw patient data are protected and not available due to the data privacy laws. The individual de-identified patient data will be made available upon request for academic research purposes from the corresponding author (zlyyliuzhenzhen0800@zzu.edu.cn) for a minimum of 10 years. The remaining data are available within the Article, Supplementary Information or Source Data file. Source data are provided with this paper.

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

## Acknowledgements
This study was supported by Jiangsu Hengrui Pharmaceuticals Co., Ltd., which provided camrelizumab free of charge. We thank all the patients, their families, and the medical staff who participated in this study. We also thank Zheng Pang, Yufen Xiang, Yuyu Du, and Jinyun Sun from Jiangsu Hengrui Pharmaceuticals Co., Ltd. for their medical writing assistance, and Yitao Wang from Jiangsu Hengrui Pharmaceuticals Co., Ltd. for his statistical support.

## Author contributions
All authors verify that this study was done per protocol and vouch for data accuracy and completeness. All authors reviewed and edited the manuscript, provided final approval to publish, and agreed to be accountable for all aspects of the article. C.W. Z.Liu., X.C., J.Q., Z.Lu., L.L. X.S. and C.Z. contributed to the design of the study; C.W. and Z.Liu. supervised and directed the study; C.W., Z.Liu., X.C. and M.Y. contributed to the development of methodology. Q.X. and H.Z. provided guidance on pathology and PD-L1 detection. C.W. and X.Y. performed data collection and analyses and manuscript drafting.

## Competing interests
The authors declare no competing interests.
