## [Peer Review File · Nature Communications]

Neoadjuvant camrelizumab plus nab-paclitaxel and epirubicin
in early triple-negative breast cancer: A single-arm phase II
trialREVIEWER COMMENTS

Reviewer #1 (Remarks to the Author): with expertise in breast cancer, therapy

Overall, this is a well written manuscript with interesting data. I have a few questions for the authors to consider. The pCR rate associated with this regimen appears similar to that with the KEYNOTE-522 regimen.

1) Given that it appears that the patients received 6 cycles of taxane-based therapy (albeit without carboplatin), What was the rate of neuropathy due to nab-paclitaxel?

2) Could the authors clarify if epirubicin was given concurrently with nab-paclitaxel or if they were given sequentially? The text seems to suggest that epirubicin was given concurrently with nab-paclitaxel.

2) Were patients with N3 disease at diagnosis enrolled in this study? If so, how many patients had N3 disease? KEYNOTE-522 did not enroll patients with N3 disease but Impassion031 enrolled patients with N3 disease.

4) In the statistical plan, it is unclear to me if the null hypothesis was a pCR rate of 30% or 48%. 30% seems a little too low as standard anthracycline-taxane chemotherapy usually yields a pCR rate of about 40%.

5) Are there data from any correlative studies that can be included in the manuscript?

Reviewer #2 (Remarks to the Author): with expertise in breast cancer, therapy

The authors present a single arm, single center clinical trial in early TNBC patients that received nab-Paclitaxel, epirubicin in combination with the ICI camrelizumab.

In summary, treatment response was measured as pCR and showed promising results comparable to the pCR rate in the much larger Keynote 522 trial. However, long term survival results are still pending. The treatment design is intriguing as it lacks cyclophosphamide and carboplatinum, but results in quite high pCR rates in this small

cohort. Interestingly, the authors report a significant difference in pCR rates regarding PD-L1 positivity.

The introduction covers the current knowledge, but some sentences should be corrected (for example line 76).

Figure 2a is not clear. The authors should explain why a patient is considered stable disease although achieving a pCR.

Figure 3 should be redrawn as it is hard to understand in the PDF version. Some balks are missing etc.

The discussion should also focus on adverse events in this trial compared to the carboplatinum containing regimens.

However, the discussion should be carefully reviewed to improve the language.

Reviewer #3 (Remarks to the Author): with expertise in breast cancer, therapy

In this monocentric, phase II study, Wang and colleagues aimed to evaluate efficacy and safety of neoadjuvant camrelizumab plus nab-paclitaxel and epirubicin in patients with early TNBC. This report provides surgical outcomes, with pCR as the study primary endpoint.

1. Lines 23-24. "For patients with high-risk early TNBC, neoadjuvant chemotherapy is the preferred treatment option". It should be noted that the 2022 ASCO Guideline Update for the treatment of high-risk, early-stage triple-negative breast cancer now recommends the use of pembrolizumab in combination with neoadjuvant chemotherapy. Therefore, it may be worth considering adding this information to the manuscript to reflect the most up-to-date treatment guidelines for high-risk early TNBC

2. Line 65. "Immune checkpoint inhibitors (ICIs) have emerged as a new potential treatment option for breast cancer". As noted above, Immune checkpoint inhibitors are already standard of care treatment for breast cancer. Therefore, I would suggest rephrasing this

statement to more accurately reflect the current state of treatment options for breast cancer patients

3. Lines 94-97. "TNBC, defined as ER and PR negative or low expression by immunohistochemistry (PR or ER staining in <10% of tumor cells irrespective of intensity) and HER2 negative by immunohistochemistry (1, 1+, or 2+) or fluorescence in situ hybridization (≤ 2.2) in core biopsies". While I agree with the authors that HR-low tumors are biologically and clinically similar to triple-negative tumors, they do not formally meet the criteria to be defined as 'triple negative' (ER and PgR <1%, HER2-negative). Additionally, the FISH 2.2 cut-off for defining HER2 status is outdated; since 2013, ASCO guidelines recommend the 2.0 cut-off. Therefore, it may be worth considering a more proper approach to defining triple negative tumor

4. Lines 221-225. The authors stated that they employed nab-paclitaxel because it did not require steroid premedication. However, they also employed epirubicin 75 mg/m² every 3 weeks, which is considered an agent with moderate emetic risk. I could not find any indication for antiemetic prophylaxis in the protocol. According to guidelines, steroids should be included for moderate emetic risk regimens. Therefore, I would like to know what kind of premedication was used

5. In Figure 2, the primary endpoint of the study, pCR, could be better highlighted. To provide more clarity and relevance, I would suggest considering the addition of a separate bar plot within the figure that focuses specifically on the pCR outcome. This would allow for a more straightforward interpretation of the results related to the study's primary endpoint

6. Figure 3 appears to be of low graphical quality and may be difficult to interpret. To ensure that the results are presented clearly and accurately, I suggest considering the possibility of redoing this figure with improved graphics

7. In Figure 3, there appears to be a discrepancy in the number of patients included in the ER-low (1-10%) tumors subgroup and the PD-L1 negative tumors subgroup when compared to Table 1. Specifically, the ER-low subgroup appears to include 13 patients in the figure but

only 10 patients in the table, while the PD-L1 negative subgroup appears to include 22 patients in the figure but 23 patients in the table. I would suggest that the authors address these discrepancies in order to clarify the exact number of patients included in each subgroup

8. Ref 24 and 25 are the same study

9. I would recommend that the authors consider elaborating the discussion section to present a more comprehensive and updated review of the current state of knowledge on neoadjuvant immunotherapy for triple negative breast cancer. This could entail discussing recent studies and trials on this subject, as well as possible avenues for future research. For instance, the authors may consider discussing the potential management options for patients who have achieved pathological complete response after neoadjuvant immunotherapy, or the appropriate course of action for those with residual disease. This would provide a better perspective and context for the study and make it a more valuable resource for readers interested in this area.

Reviewer #4 (Remarks to the Author): with expertise in clinical trial study design, biostatistics

Review of the Paper “Neoadjuvant camrelizumab plus nab-paclitaxel and epirubicin for early triple negative breast cancer: A single-arm, open-label phase II trial “

In this paper, the results from a single-arm open-label phase II study of treatment-naïve early triple-negative breast cancer (TNBC) patients treated with camrelizumab+nab-paclitaxel+epirubicin are reported. The primary endpoint was pathological complete response defined as the percentage of patients with no evidence of viable tumor cells in both breasts and no metastasis in lymph nodes (ypT0/Tis ypN0). The secondary endpoints included safety, objective response rate, event-free- survival, disease-free survival, and distant disease-free survival. However, the survival endpoints were not reported in this manuscript.

The results of the study seem promising. However, without a comparator group, it is difficult to assess the improvement over existing therapies. Below are my comments about the statistical methods and analyses.

1. It is not clear from the description of the sample size whether the improvement of pCR to 55% was assumed from 30% (an increase of 25%) or from 48% (an improvement of only 7%). Assuming it is the former (because otherwise, the required sample size would have been much higher), my calculation gives a total of $n = 31$ without dropout resulting in 35 required with 10% dropout.

2. What was the justification for the 10% drop-out rate?

3. The subgroup analysis figure (Figure 3) is not readable. Some subgroup confidence interval lines are not visible. What is the shaded area in the middle?

4. Please remove the nominal p-value for the PD-L1 from the text as this study was not designed to assess this difference. Instead, report the confidence interval of the difference between the response rates from PD-L1+ and PD-L1- patients.

5. What is on the Y-axis of Figure 2A? It would be useful for readers to see further explanation of this figure in the results section and in the caption.

Other comments:

The manuscript contains grammatical errors. Needs to be proofread carefully. Here are a few examples:

Lines 31-32: "There were 35 patients who achieved to treatment response" does not make sense.

Line 34: "...showed a promising" should be "...showed promising"

Line 158: "All statistical analysis was performed..." should be "All statistical analyses were performed..."

Response letter

Response to the reviewer's comments:

Reviewer #1

Overall, this is a well written manuscript with interesting data. I have a few questions for the authors to consider. The pCR rate associated with this regimen appears similar to that with the KEYNOTE-522 regimen.

Comment 1: *Given that it appears that the patients received 6 cycles of taxane-based therapy (albeit without carboplatin), What was the rate of neuropathy due to nab-paclitaxel?*

Response: Thanks for your comments and suggestions. We have double-checked the data that peripheral neuropathy occurred in 21 (53.8%) patients in this study. All the symptoms were grade 1 or 2 and were resolved after symptomatic treatment. We have

previously misclassified it as hand-foot syndrome and have made the corresponding correction, as shown in the revised Table 1. Thanks again for your time and constructive comments.

Comment 2: *Could the authors clarify if epirubicin was given concurrently with nab-paclitaxel or if they were given sequentially? The text seems to suggest that epirubicin was given concurrently with nab-paclitaxel.*

Response: Thanks for your comments. Epirubicin (75 mg/m² on day 1) was given concurrently with nab-paclitaxel (125 mg/m² on days 1, 8, and 15), as mentioned in the (Methods) Treatments section, according to the recommendation of the NCCN Guidelines for Breast Cancer version 3. 2019 and the Guidelines of Chinese society of clinical Oncology (CSCO) on diagnosis and treatment of breast cancer (2019 version). We have made some revision of our statement and hope it will meet with your approval.

The following is added in the manuscript:

“Chemotherapy consisted of nab-paclitaxel at 125 mg/m² on days 1, 8, and 15 and concurrent epirubicin at 75 mg/m² on day 1 every three-week, according to the recommendation of the NCCN Guidelines for Breast Cancer version 3. 2019 and the Guidelines of Chinese society of clinical Oncology (CSCO) on diagnosis treatment of breast cancer (2019 version)” **(Method: Page 16 Lines 335-Page 16 Line 337)**

Comment 3: *Were patients with N3 disease at diagnosis enrolled in this study? If so,*

how many patients had N3 disease? KEYNOTE-522 did not enroll patients with N3 disease but Impassion031 enrolled patients with N3 disease.

Response: Thanks for your comments. Three (7.7%) patients with N3 disease were enrolled in this study, as shown in Table 1. Among them, none of the 3 patients achieved a pCR. No patients in the KEYNOTE-522, while 10 (6%) patients in the IMpassion031 had N3 disease. We have added some information in the discussion section.

The following is added in the manuscript:

“Notably, the number of patients with N3 disease enrolled varied among the studies mentioned above (KEYNOTE-522: 0/784 [0%]; IMpassion031: 10/165 [6%]; GeparNuevo: 2/88 [2%]; NeoTRIP: 19/138 [14%]). None of the three (7.7%) patients with N3 disease enrolled achieved a pCR in this study. Given the small number of patients, future studies are still needed to clarify whether this subgroup of patients can benefit from the treatment.” **(Discussion: Page 9 Lines 181-186)**

Comment 4: *In the statistical plan, it is unclear to me if the null hypothesis was a pCR rate of 30% or 48%. 30% seems a little too low as standard anthracycline-taxane chemotherapy usually yields a pCR rate of about 40%.*

Response: Thanks for your comments. The null hypothesis was a pCR rate of 30% according to the previous studies and our clinical experience at the time of study design. The GeparSepto—GBG 69 trial of neoadjuvant taxanes (four 3-week cycles) in

combination with sequential epirubicin and cyclophosphamide (four 3-week cycles) reported a pCR rate of 38.4% with nab-paclitaxel and 29.0% with solvent-based paclitaxel [1]. A systemic review and meta-analysis reported a pCR rate of 33.3% with neoadjuvant platinum-free chemotherapy [2]. In this study, neoadjuvant camrelizumab was given along with concurrent nab-paclitaxel and epirubicin chemotherapy (without cyclophosphamide) for six 3-week cycles. Put all these together, the null hypothesis of a pCR of 30% was adopted for neoadjuvant concurrent anthracycline and nab-paclitaxel chemotherapy in this study. We have declared it (“The null hypothesis of a pCR rate of 30% was adopted for neoadjuvant concurrent anthracycline and nab-paclitaxel chemotherapy in this study”) in the statistical plan.

The following is added in the manuscript:

“The null hypothesis of a pCR rate of 30% was adopted for neoadjuvant concurrent anthracycline and nab-paclitaxel chemotherapy in this study.” **(Method: Page 18 Lines 371-372)**

[1] Untch, Michael et al. “Nab-paclitaxel versus solvent-based paclitaxel in neoadjuvant chemotherapy for early breast cancer (GeparSepto-GBG 69): a randomised, phase 3 trial.” *The Lancet. Oncology* vol. 17,3 (2016): 345-356. doi:10.1016/S1470-2045(15)00542-2.

[2] Poggio, F et al. “Platinum-based neoadjuvant chemotherapy in triple-negative breast cancer: a systematic review and meta-analysis.” *Annals of oncology: official journal of the European Society for Medical Oncology* vol. 29,7 (2018): 1497-1508. doi:10.1093/annonc/mdy127.

Comment 5: *Are there data from any correlative studies that can be included in the manuscript?*

Response: Thanks for your comments and suggestions. We totally agree that data from the correlative studies had better be included in the manuscript. Unfortunately, due to the limited samples with respect baseline biopsy, we only performed PD-L1 expression analysis. We have added some limitation in the main text.

The following is added in the manuscript:

“Besides, correlative studies of stromal tumor-infiltrating lymphocytes and tumor mutational burden were not performed in this study.” (**Discussion: Page 14 Lines 271-272**)

Reviewer #2

The authors present a single arm, single center clinical trial in early TNBC patients that received nab-Paclitaxel, epirubicin in combination with the ICI camrelizumab.

In summary, treatment response was measured as pCR and showed promising results comparable to the pCR rate in the much larger Keynote 522 trial. However, long term survival results are still pending. The treatment design is intriguing as it lacks cyclophosphamide and carboplatinum, but results in quite high pCR rates in this small cohort. Interestingly, the authors report a significant difference in pCR rates regarding PD-L1 positivity.

Comment 1: *The introduction covers the current knowledge, but some sentences should be corrected (for example line 76).*

Response: Thanks for comments and suggestion. We have corrected the statement.

The following is added in the manuscript:

“Anthracycline combined with taxanes is a common chemotherapy regimen for the treatment of TNBC. However, there is a lack of research on the combination of immune checkpoint inhibitors to anthracycline and taxanes in the neoadjuvant setting.”

(Introduction: Page 5 Lines 86-88)

Comment 2: *Figure 2a is not clear. The authors should explain why a patient is considered stable disease although achieving a pCR.*

Response: Thanks for your constructive comments and suggestions. We have added some detailed information about Figure 2a to make it clearer, as shown in the revised figure legends. Besides, we have double-checked the data that the patient did have a stable disease while achieved a pCR, as shown in the Figure below. Two similar cases with enlarged tumors on presurgical CT scans (RECIST-defined SD) while had minimal or no residual tumors in the surgical specimen were also noted in a previous study of neoadjuvant nivolumab in resectable lung cancer [3]. The discordance between radiological response and pathological response has long been reported [4,5]. One possible reason for this phenomenon is that conventional radiological response

assessment (e.g. CT) may provide only a macroscopic evaluation of the tumor size changes that are confounded by inflammatory or fibrotic changes [6]. Immunotherapy exerts a cell-based cytotoxic effect through harnessing host immune cells to infiltrate tumors. The increased tumor size on presurgical CT scans following neoadjuvant immunotherapy may be because of immune-cell infiltration into the tumor, rather than true tumor growth [3]. Besides, MRI has been reported to have a tendency of overestimating tumor size and number of foci [7]. We have described the case in the results and sincerely hope our response will meet with your approval.

The following is added in the manuscript:

“Figure 2 Clinical and pathological response following the neoadjuvant camrelizumab plus chemotherapy. (A) The waterfall plot showing the best percentage changes in tumor size of target lesions from baseline. Green bars indicate patients having the best overall response of partial response, and yellow bars indicate those having stable disease. The red boxes above indicate patients achieving a pathological complete response (pCR), while grey boxes indicate those without. The grey horizontal line indicates a 30% decrease from the baseline. (B) The swimmer plot showing the individual patient's journey from treatment initiation to surgery. The length of the bar indicates the duration from treatment initiation to surgery, and the colors indicate the pathological tumor response (light pink: pCR, baby blue: non-pCR). Tumor response at each assessment is shown as a red circle (partial response), grey cross (stable disease), or purple triangle (progressive disease). The reason for the end of the

treatment is shown as a grey triangle (discontinuation due to disease progression), square (discontinuation due to toxicity), or circle (completion). Green lines indicate the duration from the end of the treatment to surgery.” **(Figure legends: Page 26 Lines 531-544)**

“Interestingly, one patient had a radiological stable disease while achieving a pCR following neoadjuvant therapy with camrelizumab plus chemotherapy” **(Results: Page 6 Line 115-Page 7 Line 117)**

[3] Forde, Patrick M., et al. "Neoadjuvant PD-1 blockade in resectable lung cancer." *New England Journal of Medicine* 378.21 (2018): 1976-1986.

Immunotherapy eradicates tumor cells by harnessing host immune cells to infiltrate tumors and therefore exerting a cell-based cytotoxic effect.

[4] Falcone, Veronica et al. “Correlation Between Preoperative Radiological and Postoperative Pathological Tumor Size in Patients with HER2+ Breast Cancer After Neoadjuvant Chemotherapy Plus Trastuzumab and Pertuzumab.” *Clinical breast cancer* vol. 22,2 (2022): 149-160. doi:10.1016/j.clbc.2021.05.017.

[5] Nie, Runcong et al. “Predictive value of radiological response, pathological response and relapse-free survival for overall survival in neoadjuvant immunotherapy trials: pooled analysis of 29 clinical trials.” *European journal of cancer* (Oxford, England: 1990) vol. 186 (2023): 211-221. doi:10.1016/j.ejca.2023.03.010.

[6] William, William N Jr et al. “Computed tomography RECIST assessment of histopathologic

response and prediction of survival in patients with resectable non-small-cell lung cancer after neoadjuvant chemotherapy.” Journal of thoracic oncology : official publication of the International Association for the Study of Lung Cancer vol. 8,2 (2013): 222-8. doi:10.1097/JTO.0b013e3182774108.

[7] Di Pasquale Guadalupe, Lorena et al. “Tumor size and focality in breast carcinoma: Analysis of concordance between radiological imaging modalities and pathological examination at a cancer center.” Annals of diagnostic pathology vol. 48 (2020): 151601. doi:10.1016/j.anndiagpath.2020.151601.

Comment 3: *Figure 3 should be redrawn as it is hard to understand in the PDF version.*

Some balks are missing etc.

Response: Sorry for our mistake and any inconvenience caused. We have redrawn the Figure 3 and have added some description to make it clearer, as shown in the newly uploaded Figure 3 and revised figure legend, sincerely hoping it will meet with your approval.

The following is added in the manuscript:

“Figure 3 Subgroup analysis of pCR. The black vertical dotted line indicates the pathological complete response rate in the overall population, and the baby blue area shows the corresponding 95% confidential interval (CI). The black squares indicate the pCR for each subgroup of patients, while the length of black horizontal lines indicates the corresponding 95% CI.” (Figure legends: Page 26 Lines 545-549)

Comment 4: *The discussion should also focus on adverse events in this trial compared to the carboplatin containing regimens.*

Response: Thanks for your comments and suggestions. We have added some discussion about adverse events compared to the carboplatin-containing regimens

and sincerely hope it will meet with your approval.

The following is added in the manuscript:

“Adverse events observed in this study were generally consistent with the known safety profiles of platinum-free neoadjuvant chemotherapy for patients with early TNBC and with the known safety profiles camrelizumab in early or metastatic TNBC ^{10-12,34}.”

(Discussion: Page 12 Lines 232-234)

“Platinum-free neoadjuvant chemotherapy has been associated with a lower incidence of grade 3 or 4 hematological adverse events (e.g., neutropenia, anemia, and thrombocytopenia) when compared with platinum-based one in a previous meta-analysis ³⁵. Our study showed a comparatively lower incidence of grade 3 or 4 neutropenia/decreased neutrophil count (41.0%) when compared with previous studies of immunotherapy in combination with platinum-based neoadjuvant chemotherapy (KEYNOTE-522: 53.3%; NeoTRIP: 49%) ^{8,21}. Meanwhile, we did not find any signals of less hematological toxicity regarding anemia, leukopenia, or thrombocytopenia. However, this indirect cross-trial comparison should be interpreted with caution.” **(Discussion: Page 12 Lines 237-246)**

Comment 5: *However, the discussion should be carefully reviewed to improve the language.*

Response: Thanks for your comments. We have polished the language throughout

the manuscript.

Reviewer #3

In this monocentric, phase II study, Wang and colleagues aimed to evaluate efficacy and safety of neoadjuvant camrelizumab plus nab-paclitaxel and epirubicin in patients with early TNBC. This report provides surgical outcomes, with pCR as the study primary endpoint.

Comment 1: *Lines 23-24. “For patients with high-risk early TNBC, neoadjuvant chemotherapy is the preferred treatment option”. It should be noted that the 2022 ASCO Guideline Update for the treatment of high-risk, early-stage triple-negative breast cancer now recommends the use of pembrolizumab in combination with neoadjuvant chemotherapy. Therefore, it may be worth considering adding this information to the manuscript to reflect the most up-to-date treatment guidelines for high-risk early TNBC.*

Response: Thanks for your constructive comments and suggestion. In the first paragraph of the Introduction section, we mentioned the disease burden of and the role of neoadjuvant therapy (traditional chemotherapy) in TNBC, while, in the second paragraph, we mainly focused on the immunotherapy (e.g. pembrolizumab). Therefore, we have rephrased the sentence “neoadjuvant chemotherapy is the preferred treatment option” in the first paragraph, deleted the sentence “There is, therefore, an unmet need of new treatment strategies for early TNBC”, and have added some

information about neoadjuvant pembrolizumab in combination with chemotherapy in the second paragraph to better reflect the most up-to-date treatments. We sincerely hope the revision we made will meet with your approval.

The following is added in the manuscript:

“Cytotoxic chemotherapy remains the backbone of standard systemic treatment for TNBC due to the lack of specific targets. For patients with high-risk early TNBC, neoadjuvant chemotherapy remains the major choice.” **(Introduction: Page 4 Line 56-60)**

“Based on data from the KEYNOTE-522 trial, pembrolizumab combined with neoadjuvant chemotherapy has been recommended for high-risk early TNBC.” **(Introduction: Page 5 Lines 75-77)**

Comment 2: *Line 65. “Immune checkpoint inhibitors (ICIs) have emerged as a new potential treatment option for breast cancer”. As noted above, Immune checkpoint inhibitors are already standard of care treatment for breast cancer. Therefore, I would suggest rephrasing this statement to more accurately reflect the current state of treatment options for breast cancer patients.*

Response: Thanks for your comments and suggestion. We have rephrased the statement “Immune checkpoint inhibitors (ICIs) have been established as a part of the standard of care for the treatment of breast cancer” **(Introduction: Page 4 Line 70-**

Page 5 Line 72), sincerely hoping it will meet with your approval.

Comment 3: *Lines 94-97. “TNBC, defined as ER and PR negative or low expression by immunohistochemistry (PR or ER staining in <10% of tumor cells irrespective of intensity) and HER2 negative by immunohistochemistry (1, 1+, or 2+) or fluorescence in situ hybridization (≤ 2.2) in core biopsies”. While I agree with the authors that HR-low tumors are biologically and clinically similar to triple-negative tumors, they do not formally meet the criteria to be defined as 'triple negative' (ER and PgR <1%, HER2-negative). Additionally, the FISH 2.2 cut-off for defining HER2 status is outdated; since 2013, ASCO guidelines recommend the 2.0 cut-off. Therefore, it may be worth considering a more proper approach to defining triple negative tumor*

Response: Thanks for your constructive comments. We share your concerns about the definition of TNBC in this study. We did include some patients with HR-low tumors that are biologically and clinically similar to triple-negative tumors but do not formally meet the criteria of 'triple negative' (ER and PgR <1%, HER2-negative). To avoid possible misunderstanding, we have added some information about the patient inclusion and definition of TNBC in the discussion section. Besides, we have double-checked and confirmed that the FISH 2.0 cut-off was used to define HER2 status. We are deeply sorry for the mistake and any inconvenience that might be caused, sincerely hoping our response and revision will meet with your approval.

The following is added in the manuscript:

“One thing worth noting is that a 10% cut-off was used in this study to define ER/PR negativity, that is, patients with hormone receptor (HR)-low tumors ($1\% \leq \text{ER/PR} < 10\%$) that do not formally meet the criteria of 'triple negative' (ER and PgR $< 1\%$, HER2-negative) were also enrolled. These HR-low tumors are known to be biologically and clinically similar to TNBC in terms of pCR and survival outcomes and have long been proposed to be redefined as TNBC ²²⁻²⁵. The inclusion of this group of patients seems to be encouraging. Our subgroup analysis showed promising antitumor activity in terms of pCR irrespective of ER expression level ($< 1\%$: 58.6%; 1~10%: 80.0%), suggesting the antitumor activity of neoadjuvant immunotherapy in combination with chemotherapy in patients with HR-low tumors. The ongoing phase III trials (KEYNOTE-756 and CheckMate 7FL) will help to clarify the role of the addition of ICIs (pembrolizumab and nivolumab) to neoadjuvant chemotherapy for early HR-low, HER2- cancers ^{26,27}.” **(Discussion: Page 10 Lines 187-199)**

“Another advantage of this study is that patients with HR-low tumors that are biologically and clinically similar to triple-negative tumors but do not formally meet the 'triple negative' criteria were enrolled.” **(Discussion: Page 14 Lines 265-268)**

“Eligible patients were women aged 18 years or older with an Eastern Cooperative Oncology Group (ECOG) performance status of 0~1 and histologically confirmed treatment-naïve non-metastatic (stage IIA-IIIC) TNBC, defined as ER and PR negative or low expression by immunohistochemistry (PR or ER staining in $< 10\%$ of tumor cells

irrespective of intensity) and HER2 negative by immunohistochemistry (1, 1+, or 2+) or fluorescence in situ hybridization (< 2.0) in core biopsies” (**Methods: Page 15 Line 296**)

Comment 4: *Lines 221-225. The authors stated that they employed nab-paclitaxel because it did not require steroid premedication. However, they also employed epirubicin 75 mg/m² every 3 weeks, which is considered an agent with moderate emetic risk. I could not find any indication for antiemetic prophylaxis in the protocol. According to guidelines, steroids should be included for moderate emetic risk regimens. Therefore, I would like to know what kind of premedication was used*

Response: Thanks for your comments. The 5-HT₃ RA and/or NK-1 RA were routinely administered for antiemetic prophylaxis in this study. We have added the corresponding information.

The following is added in the manuscript:

“Prophylactic antiemetics (5-HT₃ RA, NK-1 RA) were routinely administered.”

(Methods: Page 16 Lines 320-321)

Comment 5: *In Figure 2, the primary endpoint of the study, pCR, could be better highlighted. To provide more clarity and relevance, I would suggest considering the addition of a separate bar plot within the figure that focuses specifically on the pCR outcome. This would allow for a more straightforward interpretation of the results*

related to the study's primary endpoint.

Response: Thanks for your comments and suggestion. The primary endpoint of this study is pCR (ypT0/Tis ypN0). We have already added the pCR status (pCR and non-pCR) at the top of the waterfall plot. To make it clearer, we have added some description as shown in the revised figure legends (**Page 26 Lines 531-544**), sincerely hoping that it will still meet with your approval.

Comment 6: *Figure 3 appears to be of low graphical quality and may be difficult to interpret. To ensure that the results are presented clearly and accurately, I suggest considering the possibility of redoing this figure with improved graphics*

Response: We are deeply sorry for any inconvenience caused. We have redrawn the Figure 3 and have added some description as mentioned above (**Figure legends: Page 26 Lines 545-549**).

Comment 7: *In Figure 3, there appears to be a discrepancy in the number of patients*

included in the ER-low (1-10%) tumors subgroup and the PD-L1 negative tumors subgroup when compared to Table 1. Specifically, the ER-low subgroup appears to include 13 patients in the figure but only 10 patients in the table, while the PD-L1 negative subgroup appears to include 22 patients in the figure but 23 patients in the table. I would suggest that the authors address these discrepancies in order to clarify the exact number of patients included in each subgroup

Response: Thanks for your comments and sorry for our mistakes. We have double-checked the data and have made the corresponding correction (pCR: 10/23 PD-L1 negative; 17/29 ER <1%, while 8/10 ER 1~10%), as shown in the newly uploaded Figure 3. Thanks again for your time and kind work on our manuscript.

Comment 8: *Ref 24 and 25 are the same study*

Response: Thanks, and sorry for our mistake. We have double-checked all the references and have made the corresponding correction.

Comment 9: *I would recommend that the authors consider elaborating the discussion section to present a more comprehensive and updated review of the current state of knowledge on neoadjuvant immunotherapy for triple negative breast cancer. This could entail discussing recent studies and trials on this subject, as well as possible avenues for future research. For instance, the authors may consider discussing the potential management options for patients who have achieved pathological complete response after neoadjuvant immunotherapy, or the appropriate course of action for those with*

residual disease. This would provide a better perspective and context for the study and make it a more valuable resource for readers interested in this area.

Response: Thanks for your constructive comments and suggestions. We have revised the discussion section to present a more comprehensive and updated review of the current literature on neoadjuvant immunotherapy for TNBC, as shown in the revised version of the manuscript. Also, we have added some information about the challenges faced and possible avenues for future work, as well as some discussion about patients enrolled (definition of triple-negative) and potential advantages of the study, according to your suggestions and those from other reviewers. Thanks again for your time and kind work on our manuscript.

The following is added in the manuscript:

“However, the data seem a little bit higher than that of neoadjuvant durvalumab plus sequential nab-paclitaxel and anthracycline-based chemotherapy in the GeparNuevo (53.4%) trial ²⁰ and that of neoadjuvant atezolizumab plus concurrent carboplatin and nab-paclitaxel in the NeoTRIP (48.6%) trial ²¹.” **(Discussion: Page 9 Lines 176-179)**

“However, the conflicting results of the significant effect of PD-L1 positivity on the prediction of response to neoadjuvant durvalumab or atezolizumab in combination with chemotherapy were observed in the GeparNuevo and NeoTRIP trials ^{20,21}. What’s more, PD-L1 expression was identified as the most significant factor for predicting pCR in the NeoTRIP trial.” **(Discussion: Page 11 Lines 215-221)**

“Given the high costs and potential immune-related toxicities, identifying biomarkers for more precise patient selection is of significant importance.” **(Discussion: Page 12 Lines 227-229)**

“Additionally, the assessment of long-term survival data and safety profiles is crucial when evaluating a potentially curative treatment. The contribution of adjuvant therapy with ICIs to treatment efficacy remains to be established, as well as the optimal treatment to be offered in cases of residual disease.” **(Discussion: Page 14 Lines 272-276)**

Reviewer #4

Review of the Paper “Neoadjuvant camrelizumab plus nab-paclitaxel and epirubicin for early triple negative breast cancer: A single-arm, open-label phase II trial”

In this paper, the results from a single-arm open-label phase II study of treatment-naïve early triple-negative breast cancer (TNBC) patients treated with camrelizumab+nab-paclitaxel+epirubicin are reported. The primary endpoint was pathological complete response defined as the percentage of patients with no evidence of viable tumor cells in both breasts and no metastasis in lymph nodes (ypT0/Tis ypN0). The secondary endpoints included safety, objective response rate, event-free- survival, disease-free survival, and distant disease-free survival. However, the survival endpoints were not reported in this manuscript.

The results of the study seem promising. However, without a comparator group, it is difficult to assess the improvement over existing therapies. Below are my comments about the statistical methods and analyses.

Comment 1: It is not clear from the description of the sample size whether the improvement of pCR to 55% was assumed from 30% (an increase of 25%) or from 48% (an improvement of only 7%). Assuming it is the former (because otherwise, the required sample size would have been much higher), my calculation gives a total of $n = 31$ without dropout resulting in 35 required with 10% dropout.

Response: Thanks for your constructive comments. The improvement in pCR from 30% to 55% was assumed in this study. We have added the corresponding information in the statistical analysis section. Besides, we have double-checked the sample size estimation procedure (PASS version 15.0.3) as shown below and had made some correction accordingly, sincerely hoping it will meet with your approval.

Two-Stage Phase II Clinical Trials

Possible Designs For $P_0=0.300$, $P_1=0.550$, $\alpha=0.050$, $\beta=0.200$

N1	R1	PET	N	R	Ave N	Alpha	Beta	Constraints Satisfied
25	11	0.000	25	11	25.00	0.044	0.183	Single Stage
9	2	0.463	25	11	17.59	0.042	0.199	Minimax
9	3	0.730	35	14	16.03	0.049	0.191	Optimum

The following is added in the manuscript:

“We used Simon’s two-stage design. Previous studies of neoadjuvant anthracycline-taxane-based chemotherapy showed a pCR rate of 30%~40% in TNBC^{37-40 18,35}. The null hypothesis of a pCR rate of 30% was adopted for neoadjuvant concurrent

anthracycline and nab-paclitaxel chemotherapy in this study. Assuming a pCR rate of 55% would be achieved following neoadjuvant camrelizumab plus anthracycline and nab-paclitaxel chemotherapy, a total of 39 patients (including nine in the first stage) were required when considering a one-sided α of 5%, a power of 80%, and a drop-out rate of 10%. If at least four of nine evaluable patients achieved a pCR in the first stage, the trial would proceed to the next stage, and recruitment would be continued until a total of 39 patients were enrolled. The treatment would be considered promising if 15 or more out of 35 evaluable patients achieved a pCR.” **(Method: Page 18 Line 351-Page 19 Line 360)**

Comment 2: *What was the justification for the 10% drop-out rate?*

Response: Thanks for your constructive comments. The sample size was estimated in this study on the basis of patients evaluable for pathological tumor response. The 10% drop-out rate was assumed based on our own clinical experience with reference to data from the BrightNess trial, in which the sample size was calculated after accommodation of an anticipated 10% dropout [8]. Additionally, this estimation is generally consistent with the data reported for neoadjuvant immunotherapy plus chemotherapy in TNBC. In the IMpassion031 trial, 11 of 165 (7%) patients in the atezolizumab plus chemotherapy group and 15 of 168 (9%) patients in the placebo plus chemotherapy group discontinued treatments without surgery [9]. We want to express our great appreciation again for your time and kind work on our manuscript and sincerely hope our response will still meet with your approval.

[8] Loibl, Sibylle et al. "Addition of the PARP inhibitor veliparib plus carboplatin or carboplatin alone to standard neoadjuvant chemotherapy in triple-negative breast cancer (BrighTNess): a randomised, phase 3 trial." *The Lancet. Oncology* vol. 19,4 (2018): 497-509. doi:10.1016/S1470-2045(18)30111-6.

[9] Mittendorf, Elizabeth A et al. "Neoadjuvant atezolizumab in combination with sequential nab-paclitaxel and anthracycline-based chemotherapy versus placebo and chemotherapy in patients with early-stage triple-negative breast cancer (IMpassion031): a randomised, double-blind, phase 3 trial." *Lancet (London, England)* vol. 396,10257 (2020): 1090-1100. doi:10.1016/S0140-6736(20)31953-X.

Comment 3: *The subgroup analysis figure (Figure 3) is not readable. Some subgroup confidence interval lines are not visible. What is the shaded area in the middle?*

Response: We are deeply sorry for any inconvenience caused. We have redrawn the figure and added some description as mentioned above (**Figure legends: Page 26 Lines 545-549**).

Comment 4: *Please remove the nominal p-value for the PD-L1 from the text as this study was not designed to assess this difference. Instead, report the confidence interval of the difference between the response rates from PD-L1+ and PD-L1- patients.*

Response: Thanks for your comment and suggestion. We have added the confidence interval of the difference (48.2%, 95% CI: 22.6%, 73.8%) according to the your

suggestion.

The following is added in the manuscript:

“The percentage of patients with the pCR was 91.7% (11/12) in PD-L1-positive patients, while was 40.9% (9/22) in PD-L1-negative patients (difference: 48.2%, 95% CI: 22.6%, 73.8%).” **(Results: Page 7 Line 132)**

Comment 5: *What is on the Y-axis of Figure 2A? It would be useful for readers to see further explanation of this figure in the results section and in the caption.*

Response: Thanks. The Y-axis of Figure 2A is “Best percentage change from baseline in the sum of target lesion size”. We have added some description mentioned above **(Figure legends: Page 26 Lines 531-544)**.

Other comments:

The manuscript contains grammatical errors. Needs to be proofread carefully. Here are a few examples:

Lines 31-32: “There were 35 patients who achieved to treatment response” does not make sense.

Line 34: “...showed a promising” should be “...showed promising”

Line 158: “All statistical analysis was performed...” should be “All statistical analyses were performed...”

Response: Thanks for your comments. We have made the corresponding revision and

have *proofread the manuscript according to your suggestions.*

We have tried our best to improve the manuscript and made some changes in the manuscript. These changes will not influence the content and framework of the paper.

We appreciate for Reviewers' warm work earnestly, and hope that the correction will meet with your approval.

Once again, thank you very much for your comments and suggestions.

REVIEWERS' COMMENTS

Reviewer #1 (Remarks to the Author):

I have no further comments

Reviewer #2 (Remarks to the Author):

The authors have addressed my comments sufficiently. However, in figure 3 there is a spelling mistake and I wonder if it is possible to replace hand-foot syndrome with peripheral neuropathy. These are two separate side effects which have their own diagnostic criteria. How can that be miss-interpreted by the treating physician?

Reviewer #3 (Remarks to the Author):

The authors have adequately addressed my comments and suggestions to improve the quality of the manuscript. I have also read the corrections made in response to the comments of the other reviewers, and I believe the authors have done a good job. I have no further considerations to make. Thank you.

Luca Licata, MD

Breast Cancer Group

Department of Medical Oncology

San Raffaele Hospital, Milan (Italy)

Reviewer #4 (Remarks to the Author):

Reviewers have clarified my comments and addressed the concerns in the revision. No further comments.

Response letter

Response to the reviewer's comments:

Reviewer #1:

I have no further comments

Response: Thanks again for your time and kind work on our manuscript.

Reviewer #2:

The authors have addressed my comments sufficiently. However, in figure 3 there is a spelling mistake and I wonder if it is possible to replace hand-foot syndrome with peripheral neuropathy. These are two separate side effects which have their own diagnostic criteria. How can that be miss-interpreted by the treating physician?

Response: Thanks for your further comments and suggestions. We have corrected the spelling mistake ('positive') in the Figure 3. As for the adverse events, we have double-checked the original data on diary cards of each patient in the first-round of

revision. All these side effects were mild (grade I or II), and the major complaints of the patients were numbness and tingling in the hands and/or feet (as shown in the Table 1 below), without typical skin changes (e.g., peeling, blisters, bleeding, fissures, edema, or hyperkeratosis). After careful consideration, the term peripheral neuropathy was used instead of the hand-foot syndrome. We are deeply sorry for the mistakes and inconvenience caused, sincerely hope our response will meet with your approval. Thanks again for your time and kind work on our manuscript.

Table 1 Major complaints of the patients developing peripheral neuropathy

Major complaints	Number of patients
Hands and feet numbness	10
Hands numbness	4
Feet numbness	2
Fingertips numbness	2
Fingers numbness and tingling	1
Hands and forearms numbness	1
Thumbs, lower legs, and feet numbness	1

Reviewer #3:

The authors have adequately addressed my comments and suggestions to improve the quality of the manuscript. I have also read the corrections made in response to the comments of the other reviewers, and I believe the authors have done a good job.

I have no further considerations to make. Thank you.

Response: Thanks again for your time and kind work on our manuscript.

Reviewer #4:

Reviewers have clarified my comments and addressed the concerns in the revision.

No further comments.

Response: Thanks again for your time and kind work on our manuscript.

We would likely to appreciate your warm work again earnestly, and hope that our responses will meet with your approval. Once again, thank you very much for your comments and suggestions.